# Relative Entropy and Minimum-Variance Pricing Kernel in Asset Pricing Model Evaluation

**DOI:** 10.3390/e22070721

**Published:** 2020-06-30

**Authors:** Javier Rojo-Suárez, Ana Belén Alonso-Conde

**Affiliations:** Department of Business Administration, Rey Juan Carlos University, 28032 Madrid, Spain; ana.alonso@urjc.es

**Keywords:** pricing kernel, factor-mimicking portfolio, relative entropy, Kullback-Leibler divergence, Hansen-Jagannathan bound, CAPM, Fama-French model, consumption-CAPM

## Abstract

Recent literature shows that many testing procedures used to evaluate asset pricing models result in spurious rejection probabilities. Model misspecification, the strong factor structure of test assets, or skewed test statistics largely explain this. In this paper we use the relative entropy of pricing kernels to provide an alternative framework for testing asset pricing models. Building on the fact that the law of one price guarantees the existence of a valid pricing kernel, we study the relationship between the mean-variance efficiency of a model’s factor-mimicking portfolio, as measured by the cross-sectional generalized least squares (GLS) $R^{2}$ statistic, and the relative entropy of the pricing kernel, as determined by the Kullback–Leibler divergence. In this regard, we suggest an entropy-based decomposition that accurately captures the divergence between the factor-mimicking portfolio and the minimum-variance pricing kernel resulting from the Hansen-Jagannathan bound. Our results show that, although GLS $R^{2}$ statistics and relative entropy are strongly correlated, the relative entropy approach allows us to explicitly decompose the explanatory power of the model into two components, namely, the relative entropy of the pricing kernel and that corresponding to its correlation with asset returns. This makes the relative entropy a versatile tool for designing robust tests in asset pricing.

## 1. Introduction

Research on asset pricing encompasses a wide range of theories and models that seek to identify the fundamental risk factors that determine asset prices from a broad perspective. Following [1] (p. xiii), the asset pricing theory tries to understand the prices of claims to uncertain payments by accounting for both the delay in time and the risk of future payments. Since asset prices in conjunction with expected payoffs predetermine expected returns, a large part of the asset pricing literature focuses on explaining both the time-series evolution and the cross-sectional properties of expected asset returns, that is, the variation of expected returns across time and across assets, respectively. This makes asset pricing models extremely useful tools in very diverse areas, such as capital budgeting, interest rate forecasting, financial planning, accounting, or regulation [2].

While the literature on the topic has generally analyzed a wide variety of assets, ranging from fixed-income securities to derivatives, and even non-publicly traded assets, research on the formation of stock prices has traditionally raised great interest, given its direct applicability for developing new trading strategies in a market that has historically provided returns significantly higher than others, such as the bond market [3] (p. xix). In this regard, although most empirical work on asset pricing has typically used single-factor or multifactor models to test the validity of different asset pricing models, currently the stochastic discount factor (hereafter, SDF) or pricing kernel model is the dominant approach in contemporary asset pricing research, not just for stocks but for any asset class [4] (p. 83). This model assumes that the price of every asset is given by the expectation conditional on time-*t* information of the product of the pricing kernel and the asset payoff. In this framework, the pricing kernel represents the ratio of price to probability for all states of nature or marginal rate of substitution, meaning that it can be perfectly used as a valid risk factor in a single-factor asset pricing model, in order to determine expected returns [1] (pp. 100–101).

Regarding model evaluation techniques, most cross-sectional model testing procedures focus on studying the size and/or variability of pricing errors, which are determined for each asset as the difference between the expected return—i.e., the average return in sample—and the result provided by the model. Such testing procedures usually come in the form of a quadratic test statistic determined as a function of pricing errors, as is the case of the *J*-test for overidentifying restrictions [5], the Gibbons-Ross-Shanken (GRS) test statistic [6], the Hansen-Jagannathan distance [7], or the cross-sectional $R^{2}$ statistic. However, recent evidence highlights several weaknesses for these testing procedures. Reference [1] (pp. 215–216) shows that large standard errors rather than small pricing errors can lead the *J*-test to under-reject asset pricing models. The GRS test statistic and the Hansen-Jagannathan distance share the same problem. Similarly, the ordinary least squares (OLS) $R^{2}$ statistic usually constitutes a low hurdle in examining model success, albeit the generalized least squares (GLS) $R^{2}$ statistic allows for a much more rigorous model evaluation [8].

In this paper, we adapt the approach pioneered by [9], suggesting two entropy-based indicators that are strongly correlated with asset pricing model performance, in order to overcome the problems tied to classic model evaluation tests in cross-sectional analysis. This approach may constitute a starting point for developing new testing procedures useful in asset pricing, exploiting the high potential of non-quadratic distances in model assessment [10]. Building on the fact that under very weak conditions imposed on the behavior of financial markets—in particular, the law of one price—there is at least a valid pricing kernel that prices all assets perfectly in sample [11,12,13], we use the minimum-variance pricing kernel, as defined by [14] (hereafter, HJ pricing kernel or Hansen-Jagannathan SDF), to split the explanatory power of any asset pricing model into two components, namely: (i) the relative entropy between the HJ pricing kernel and the model’s SDF-mimicking portfolio, and (ii) the relative entropy between the correlation of asset payoffs with the HJ pricing kernel and the correlation of asset payoffs with the model’s SDF-mimicking portfolio. For this purpose, we use the relative entropy resulting from the Kullback–Leibler information criterion. In order to study the relationship between our approach and a robust model evaluation test statistic, we explicitly relate the results of our model to those provided by the GLS $R^{2}$ statistic, as it constitutes a powerful measure of model performance [8].

This paper contributes to the literature that deals with specific issues on model evaluation techniques used in asset pricing [8,15,16], and to the literature that relates entropy to asset pricing theory [9,17,18,19,20]. To the best of our knowledge, this is the first study to explicitly decompose the relative entropy of pricing kernels into the specific components referred to above, relating them to the GLS $R^{2}$ statistic using a factor-mimicking portfolio approach.

Regarding issues concerning model evaluation tests in asset pricing, [8] provide a comprehensive analysis of the subject. The authors emphasize three main drawbacks of these methodologies. First, test assets that are typically used in asset pricing—namely, portfolios of stocks sorted by size and book-to-market equity (hereafter, BE/ME)—exhibit a strong factor structure that makes the models perform artificially well in sample when model factors are correlated with size and BE/ME portfolios but not with the residuals of the size-BE/ME portfolios. Second, while the cross-sectional OLS $R^{2}$ statistic is one of the most widely used tools for evaluating model performance, it has little connection to the factors’ proximity to the minimum-variance frontier in the mean-variance space of returns. This is a critical issue, since any return on the minimum-variance frontier carries all pricing information about payoffs in the payoff space, providing zero pricing errors when used as a factor in a single-factor model [1] (p. 19). As noted above, [8] argue that this problem is significantly mitigated by the cross-sectional GLS $R^{2}$ statistic. Following [21], the authors highlight that the GLS $R^{2}$ statistic is entirely determined by the factor’s proximity to the minimum-variance frontier, which makes it more suitable than the OLS $R^{2}$ statistic to evaluate the performance of asset pricing models. Third, [8] explain that asset pricing test statistics are frequently biased and skewed, leading to sampling issues that exacerbate the problems mentioned above and reduce the reliability of *p*-values and standard errors.

On a purely economic basis, it is worth mentioning that the vast majority of factor asset pricing models constitute special cases of the SDF model. Indeed, any single-factor or multifactor asset pricing model can be easily transformed into a SDF model simply by carrying out a linear transformation on model factors [1] (pp. 106–108). In fact, it is possible to determine a linear combination of test assets—that is, an asset portfolio—which exactly replicates the results provided by the pricing kernel. This factor-mimicking portfolio constitutes a useful tool in different situations, for example, when there is some measurement error in the set of economic variables driving the pricing kernel [1] (pp. 125–126). In this paper, we use that factor-mimicking portfolio to synthesize all model factors into one single explanatory variable, in order to determine its relative entropy with respect to the HJ pricing kernel. Importantly, the HJ pricing kernel is the minimum-variance mimicking portfolio that allows the SDF model to price perfectly all asset payoffs in the payoff space [14], among those resulting from the linear combination of a constant—the risk-free rate—and *N* risky payoffs.

Our results show that our entropy-based approach allows sorting factor-based asset pricing models in a similar manner to most model evaluation statistics. Specifically, using different sets of portfolios comprising all stocks traded in the Australian market, for the period from 1992 to 2018, we show that our entropy decomposition is strongly correlated with the GLS $R^{2}$ statistic, for some of the most prominent asset pricing models in the literature, namely, the capital asset pricing model (CAPM) [22,23,24], the consumption-CAPM (CCAPM) [25,26] and the Fama-French three- and five-factor models [27,28]. Indeed, our results suggest that the entropy of the pricing kernel, as defined above, explains up to 78% of the variability of the GLS $R^{2}$ statistic. Our results also reveal that, among the components into which we divide the explanatory power of the model, the relative entropy of the correlation between pricing kernels and asset payoffs is the most strongly correlated with GLS $R^{2}$ statistics, reaching a correlation of up to 72% in absolute terms. Interestingly, that component accurately accounts for the coherence of pricing kernels over time, relative to the HJ pricing kernel.

Hereafter, the paper proceeds as follows. The next section defines the models and methods under analysis. The section thereafter describes the data and the results. This is followed by a section that discusses the results. The last section concludes the paper.

## 2. Models and Methods

In this section we first describe the general mechanics of the pricing kernel model and relate it to factor models, as they constitute the most widely used representations for estimating asset pricing models. Subsequently, we describe the procedure that allows us to switch from a factor-based asset pricing model to a pricing kernel model, using the SDF-mimicking portfolio for that purpose. This allows us to comprise all the factors of the model into a single explanatory variable—the SDF-mimicking portfolio—and standardize the data frequency for all models under consideration, which is essential given the presence of low-frequency series—e.g., consumption data—among the factors considered. Finally, we use the HJ pricing kernel, which exactly satisfies the pricing function by construction, to suggest an entropy decomposition that is appropriate for evaluating the empirical performance of asset pricing models.

Assuming a complete market, where there is the same number of securities as states of nature, the law of one price—that is, the fact that every asset that provides the same payoff as another has the same price—guarantees that there is one and only one pricing kernel in the payoff space that allows all assets to be priced with no error term left in the pricing function [11,12,13]. Accordingly, the following pricing function is exactly satisfied:(1)$$p_{t}=E_{t}\left(m_{t+1}x_{t+1}\right),$$
where $p_{t}$ is the *N*-dimensional vector of asset prices at time *t*, $E_{t}\left(\cdot \right)$ is the expectation conditional on time-*t* information, $m_{t+1}$ is the pricing kernel, and $x_{t+1}$ is the *N*-dimensional vector of asset payoffs. Particularizing Expression (1) to asset returns, that is, investments with price one and payoff $R_{t+1}$:(2)$$E_{t}\left(m_{t+1}R_{t+1}\right)=1,$$
where $R_{t+1}$ is the *N*-dimensional vector of asset returns. Defining an excess return as the difference between returns, that is, a zero-cost portfolio consisting in short-selling one security to invest the proceeds in another security, then every excess return satisfies:(3)$$E_{t}\left(m_{t+1}R_{t+1}^{e}\right)=0,$$
where $R_{t+1}^{e}$ is the *N*-dimensional vector of excess returns in the payoff space.

Market completeness, together with the law of one price, ensures the existence of a single pricing kernel that satisfies Expression (1) and that lies in the payoff space. Importantly, the law of one price still guarantees the existence of a single valid pricing kernel in the payoff space when the market is incomplete—that is, when there are fewer securities than states of nature—but it does no guarantee its uniqueness. Indeed, in an incomplete market there are potentially infinite pricing kernels that satisfy Expression (1) [1] (pp. 66–67). Accordingly, we can particularize Expression (3) as follows, for the case of incomplete markets:(4)$$E_{t}\left(x_{t+1}^{*}R_{t+1}^{e}\right)=0,$$
with $x_{t+1}^{*}\in m_{t+1}$ denoting the single valid pricing kernel in the payoff space. Assuming that $m_{t+1}$ is linear in a *K*-dimensional vector of factors f (not necessarily in the payoff space), the pricing kernel can be written as follows:(5)$$m_{t+1}=a_{t}+b_{t}^{\prime }f_{t+1},$$
where $a_{t}$ and $b_{t}$ are parameters. As noted above, most factor asset pricing models can be considered special cases of the SDF model. Since most empirical work in asset pricing uses beta models rather than the pricing kernel, the Equivalence Theorem allows us to transform Expression (3) into a beta model straightforwardly [11,13,29,30]. Specifically, writing second moments in terms of covariances:(6)$$E_{t}\left(m_{t+1}R_{t+1}^{e}\right)=cov_{t}\left(m_{t+1},R_{t+1}^{e}\right)+E_{t}\left(m_{t+1}\right)E_{t}\left(R_{t+1}^{e}\right)=0,$$
or equivalently:(7)$$E_{t}\left(R_{t+1}^{e}\right)=-\frac{cov_{t}\left(m_{t+1},R_{t+1}^{e}\right)}{E_{t}\left(m_{t+1}\right)},$$
where expected excess returns are largely determined by the conditional covariance between the pricing kernel and excess returns. Multiplying and dividing Expression (7) by $\sigma_{t}^{2}\left(m_{t+1}\right)$:(8)$$E_{t}\left(R_{t+1}^{e}\right)=\left[\frac{cov_{t}\left(m_{t+1},R_{t+1}^{e}\right)}{\sigma_{t}^{2}\left(m_{t+1}\right)}\right]\left[-\frac{\sigma_{t}^{2}\left(m_{t+1}\right)}{E_{t}\left(m_{t+1}\right)}\right]=\beta_{t,m,R^{e}}\lambda_{t,m},$$
where $\lambda_{t,m}$ can be interpreted as the price of risk and $\beta_{t,m,R^{e}}$ as the risk loadings. Similarly, Expression (5) allows us to rewrite Expression (8) as follows:(9)$$E_{t}\left(R_{t+1}^{e}\right)=\beta_{t,f,R^{e}}\lambda_{t,f}$$

Although conditioning information has become increasingly popular in empirical research on asset pricing [31,32,33,34,35], most classic regression tests use unconditional rather than conditional moments. Ignoring conditioning information for the sake of simplicity, the law of iterated expectations allows us to transform the conditional SDF model into its unconditional counterpart, naturally suppressing the *t* subscripts in Expressions (3) and (5):(10)$$E\left(m_{t+1}R_{t+1}^{e}\right)=0,$$
(11)$$m_{t+1}=a+b^{\prime }f_{t+1}$$

Similarly, we can rewrite Expression (9) as follows:(12)$$E\left(R_{t+1}^{e}\right)=\beta_{f,R^{e}}\lambda_{f},$$
where $\beta_{f,R^{e}}$ is the N×K matrix of slopes of the time-series regressions of excess returns on factors f, and $\lambda_{f}$ is the vector of slopes of the cross-sectional regression of expected returns on $\beta_{f,R^{e}}$. As before, $\lambda_{f}$ can be interpreted as the prices of risk or risk premiums on factors.

Remarkably, most unconditional factor-based asset pricing models can be written in terms of Expression (12) by simply particularizing the vector of factors f to specific explanatory variables. In particular, the CAPM assumes that f=RMRF, where RMRF is the return of the value-weighted market portfolio minus the risk-free rate, while the Fama-French three- and five-factor models assume that $f^{\prime }=\left(\mathrm{RMRF}\;\mathrm{SMB}\;\mathrm{HML}\right)$ and $f^{\prime }=\left(\mathrm{RMRF}\;\mathrm{SMB}\;\mathrm{HML}\;\mathrm{RMW}\;\mathrm{CMA}\right)$, respectively, where SMB is the excess return of small minus big market value firms, HML is the excess return of the high minus low BE/ME firms, RMW is the excess return of the most profitable stocks minus the least profitable, and CMA is the excess return of firms that invest conservatively minus those that invest aggressively. Similarly, the CCAPM assumes that f=ΔC, where ΔC is the growth in per capita consumption in nondurables and services.

Once the coefficients of Expression (12) are estimated, Expressions (10) and (11) allow determining a and b as a function of $\lambda_{F}$ and model factors. Importantly, when the payoff space comprises solely excess returns, the pricing function does not identify the mean of the pricing kernel and any arbitrary value for the coefficient a will produce exactly the same fitted values. Consequently, in order to normalize the pricing kernel, we follow the common practice of setting a to one, in which case [1] (p. 107):(13)a=1
(14)$$b=-cov{\left({\tilde{f}}_{t+1},{\tilde{f}}_{t+1}^{\prime }\right)}^{-1}\lambda_{F}$$
where $\tilde{f}$ is the vector of demeaned factors. This calculation ensures that $E\left(m_{t+1}\right)=1$, which is convenient to allow comparability with the HJ pricing kernel, as defined below.

At this point, it is important to underline that, while the CAPM and the Fama-French models use traded factors as explanatory variables, the CCAPM uses a non-traded factor—per capita consumption growth—as a key determinant of asset prices. Since consumption data is generally reported both on an annual and quarterly basis, Expression (12) does not allow the CCAPM to be used at monthly or higher frequencies. However, this drawback can be overcome by determining the projection of the pricing kernel on the payoff space, that is, the SDF-mimicking portfolio $x^{*}$, which carries exactly the same pricing information as the pricing kernel. Therefore, following [1] (p. 67):(15)$$m_{t+1}=\mathrm{proj}\left(m_{t+1}|\underline{R^{e}}\right)+\varepsilon_{t+1},$$
where $\mathrm{proj}\left(m_{t+1}|\underline{R^{e}}\right)$ is the mimicking portfolio of the pricing kernel, $\underline{R^{e}}$ is the space of excess returns, and $\varepsilon_{t+1}$ is the error. Expression (15) can be rewritten as the regression of $m_{t+1}$ on the set of excess returns used as test assets. Accordingly:(16)$$m_{t+1}=w^{\prime }R_{t+1}^{e}+\varepsilon_{t+1},$$
where w is the *N*-dimensional vector of regression coefficients or portfolio weights for $x^{*}$, so that:(17)$$x_{t+1}^{*}=w^{\prime }R_{t+1}^{e}$$

Once determined $x^{*}$, Expression (4) allows us to estimate expected excess returns under the same conditions as Expression (3), but at a frequency equal to that of market data [1] (pp. 125–126).

Regardless of the performance of any specific asset pricing model, it is always possible to determine an ad hoc SDF-mimicking portfolio which prices assets by construction. That mimicking portfolio shall leave no error term in the pricing function when used in sample, although it will perform poorly out of sample, making it useless for explaining or forecasting excess returns [1] (pp. 123–125). In any case, the relative entropy between the ad hoc SDF-mimicking portfolio and any other pricing kernel allows us to explicitly determine the divergence between both discount factors, which makes the relative entropy an accurate tool to evaluate asset pricing model performance. Following [9], we set the ad hoc SDF-mimicking portfolio to the HJ pricing kernel, which is given by this expression [14]:(18)$$m_{t+1}^{*}=\frac{1}{R_{t+1}^{f}}-\frac{1}{R_{t+1}^{f}}E{\left(R_{t+1}^{e}\right)}^{\prime }\Sigma^{-1}\left[R_{t+1}^{e}-E\left(R_{t+1}^{e}\right)\right],$$
where $m_{t+1}^{*}$ is the HJ pricing kernel, $R_{t+1}^{f}$ is the risk-free rate, and Σ is the covariance matrix of excess returns. Remarkably, when the payoff space consists solely of excess returns, the risk-free rate is meaningless and can be arbitrarily chosen.

Although research on variance and entropy bounds on the discount factor focuses largely on the distribution of the pricing kernel [9,14,18], it is worth noting that the ability of an asset pricing model to correctly price excess returns depends both on the randomness of the SDF and its correlation with excess returns. Indeed, given that $R_{t+1}^{f}=1/E_{t}\left(m_{t+1}\right)$, according to Expression (7) every single excess return satisfies:(19)$$E_{t}\left(R_{t+1}^{e}\right)=-R_{t+1}^{f},\rho_{t,m,R^{e}}\sigma_{t}\left(m_{t+1}\right)\sigma_{t}\left(R_{t+1}^{e}\right),$$
where $\rho_{t,m,R^{e}}$ is the correlation at time *t* between the pricing kernel and the excess return. Expression (19) shows that the Sharpe ratio of excess returns $E_{t}\left(R_{t+1}^{e}\right)/\sigma_{t}\left(R_{t+1}^{e}\right)$ is entirely determined by $\rho_{t,m,R^{e}}$ and $\sigma_{t}\left(m_{t+1}\right)$, which means that our model evaluation procedure must consider both estimates. Accordingly, we use the Kullback-Leibler divergence to define the following performance indicators:(20)$$I_{x^{*}}\left(\pi \;\|\;\theta \right)=\int log\left(d\pi /d\theta \right)d\pi ,$$
(21)$$I_{\rho }\left(\upsilon \;\|\;\mu \right)=\int log\left(d\upsilon /d\mu \right)d\upsilon ,$$
where $I_{x^{*}}$ and $I_{\rho }$ are the Kullback-Leibler divergences for the pricing kernel and the correlation between the pricing kernel and excess returns, respectively, π and θ are the state probability measures for $m^{*}$ and $x^{*}$, respectively, and υ and μ are the state probability measures for $\rho_{m^{*},R^{e}}$ and $\rho_{x^{*},R^{e}}$, respectively.

In the next section we use Expression (12) to estimate all models under consideration and then determine their implicit pricing kernels using Expressions (11), (13) and (14). This allows us to estimate the SDF-mimicking portfolio for each model using Expressions (16) and (17), in order to switch all calculations to a monthly basis. We use the HJ pricing kernel, as determined by Expression (18), to estimate the relative entropy for both the pricing kernels and their correlations with excess returns, according to Expressions (20) and (21). Finally, we perform a regression and correlation analysis aimed at analyzing the relationship of $I_{x^{*}}$ and $I_{\rho }$ with the cross-sectional GLS $R^{2}$ statistic as a proxy for the relative efficiency of the model’s factor-mimicking portfolio [8].

## 3. Results

As noted above, in this section we study the potential of our entropy-based decomposition, as defined in the previous section, to evaluate the performance of some classic asset pricing models, namely, the CCAPM, the CAPM, and the Fama-French three- and five-factor models. Additionally, we analyze the consistency between the results provided by the model and the GLS $R^{2}$ statistic of the cross-sectional regression of expected excess returns on betas.

We evaluate our methodology on three sets of stock portfolios comprising all stocks traded in the Australian Securities Exchange, for the period from July 1992 to June 2018. We compile all market data from the Datastream database, on a monthly basis. Specifically, we compile the following data series: (i) total return index (RI series), (ii) market value (MV series), (iii) market-to-book equity (PTBV series), (iv) price-to-cash flow ratio (PC series), (v) primary SIC codes, and (vi) tax rate (WC08346 series). We use the rules suggested by [36] for excluding non-common equity securities from Datastream data. We use the three-month interest rate of the Treasury Bill for Australia, as provided by the OECD, as a proxy for the risk-free rate.

Regarding the specific portfolios that constitute our test assets, we follow the recommendations suggested by [8,37], combining portfolios sorted by different market anomalies and industry portfolios to mitigate the factor structure of test assets. Specifically, we generate the following portfolios: (i) 25 portfolios sorted by size and BE/ME (hereafter, size-BE/ME portfolios), (ii) 20 portfolios sorted by size and price-to-cash flow ratio (hereafter, size-P/CF portfolios), and (iii) 41 industry portfolios, sorted by two-digit SIC codes. We use the [27] methodology to form all portfolios. Additionally, we follow [27,28] methodology to determine RMRF, SMB, HML, RMW and CMA market factors. All data series are available in the Appendix A.

We compile all macroeconomic data series from the OECD Statistics section. Regarding consumption data, although the CCAPM requires using the per capita consumption growth in nondurables and services as a factor, the limited availability of those series for Australia in the OECD Statistics section hinders the use of that variable. In any case, [35] show that total consumption outperforms consumption in nondurables and services when used as an explanatory variable in the CCAPM. Accordingly, we estimate consumption growth using quarterly final consumption data in national currency, non-seasonally adjusted series (“Private final consumption expenditure” series, CQR measure). We use the population series for Australia, as provided by the OECD, to determine per capita consumption growth.

Table 1 shows the main summary statistics for portfolios and factors under consideration, while Table 2 shows the correlations between our test assets and the explanatory variables. Panels A and B in Table 1 show that the size effect—that is, the fact those stocks with lower market equity provide higher returns, and vice versa—works as expected, with portfolios in first size quintiles providing significantly higher returns. However, as shown in Panel A, the value effect—that is, the fact those stocks with higher BE/ME provide higher returns, and vice versa–exhibits a much more diffuse pattern, except for portfolios comprising the largest companies, which provide higher returns as the BE/ME ratio increases. Nevertheless, these counterintuitive results for the BE/ME portfolios are shared with other studies in the area [38]. Similarly, Panel B shows that P/CF quintiles do not reveal a clear pattern, with portfolios providing arbitrarily high or low returns regardless of the P/CF ratio. As usual, industry portfolios (Table 1, Panel C) provide the lowest spread between the best and the worst performing portfolios among the assets under study. Specifically, this spread amounts to 6.34% for industry portfolios, while it rises to 10.97% and 9.36% for size-BE/ME and size-P/CF portfolios, respectively.

Regarding factors, Table 1, Panel D shows that the mean excess return of the value-weighted market portfolio for the Australian Securities Exchange, RMRF, is relatively high (1.09% on a monthly basis) compared to that reported for other markets, generally ranging from 4% and 8% on an annual basis [1] (pp. 460–461). On the other hand, consumption growth statistics take normal values, with an annual mean of 1.75% and a standard deviation of 1.56%. When we determine annual consumption growth using consumption data exclusively for the fourth quarter, as suggested by [39], those statistics amount to 1.71% and 1.80%, respectively (referred to as Q4-Q4 consumption growth in Table 1).

Table 2 shows that both size-BE/ME and size-P/CF portfolios exhibit high positive correlations with RMF, SMB and RMW, while they are poorly correlated with CMA and consumption growth. On the other hand, industry portfolios are significantly less correlated with most factors, with the sole exception of RMRF. Remarkably, most portfolios under consideration are negatively correlated with HML. In any case, it is worth noting that the correlations shown in Table 2 do not determine the performance of asset pricing models under study, rather, the performance is determined by the correlation between the expected returns and the covariances (or betas) of returns and factors, as shown in Expression (9).

Table 3 summarizes the regression results provided by Expression (12) for the factor models under consideration, while Table 4 shows the estimates resulting from the pricing kernel model, with the SDF-mimicking portfolio $x^{*}$, as determined by Expression (17), as the only explanatory variable in Expression (12). For all models we use the generalized method of moments (GMM) to simultaneously estimate betas and lambdas in Expression (12). Standard errors are corrected for the cross-sectional autocorrelation of the error term and for the fact that betas are generated regressors.

Most of the asset pricing models considered in Table 3 perform as one might expect, with the Fama-French three- and five-factor models providing the best results, and the CCAPM and the CAPM performing significantly worse. Specifically, in all cases, the Fama-French five-factor model provides the lowest mean absolute error (MAE), with values ranging from 0.38% for industry portfolios (row C4 in Table 3) to 0.49% for size-P/CF portfolios (row B4). On the other hand, the CCAPM is the worst-performing model among those under study. In any case, as noted above, the lack of monthly consumption data series leads us to preliminarily estimate the CCAPM using quarterly data. This makes it necessary to divide the MAEs provided by the CCAPM by three to allow for comparison with the other models. Thus, while the CCAPM provides quarterly MAEs of 8.64%, 6.26%, and 1.63%, for size-BE/ME, size-P/CF, and industry portfolios, respectively, their monthly counterparts drop to 2.88%, 2.09%, and 0.54%. As mentioned above, the SDF-mimicking portfolio implicit in the CCAPM allows us to directly determine monthly estimates, as shown in Table 4.

The Fama-French five-factor model also provides the highest OLS $R^{2}$ statistics among the models studied in Table 3, with values exceeding 90% for size-BE/ME and size-P/CF portfolios. In contrast, the CCAPM and the CAPM provide OLS $R^{2}$ statistics of less than 50% in virtually all cases. Interestingly, the CCAPM delivers a trivial OLS $R^{2}$ statistic of 3.9% for size-BE/ME portfolios (row A1), while clearly outperforming the CAPM for industry portfolios, with an OLS $R^{2}$ statistic of 62.4% (row C1).

Notably, the *J*-test for over-identifying restrictions rejects most models considered in Table 3. Nevertheless, the *J*-test fails to reject Fama-French models in some cases, specifically for rows B4, C3 and C4. Surprisingly, the *J*-test does not reject the CCAPM for size-P/CF portfolios (row B1) despite its poor performance. In any case, it is worth mentioning that it is largely the high variances and covariances of GMM moments, rather than small pricing errors, that make the *J*-test not reject this model.

As shown in Table 3, the vast majority of risk premiums are not statistically significant, with the sole exception of SMB and, for some models, RMRF and consumption growth coefficients. Additionally, most risk premiums are far from the mean of factors (see Table 1, Panel D), meaning that, in general, the models are not well-behaved. In any case, the risk premiums on RMRF, SMB, and HML for the Fama-French five-factor model are relatively close to their means, for most portfolios under consideration.

Table 4 shows that the SDF-mimicking portfolio provides exactly the same results as the factor models shown in Table 3 for most performance indicators. In particular, the MAEs and OLS $R^{2}$ statistics shown in Table 4 are fully in line with those in Table 3, meaning that the mimicking portfolio of the pricing kernel does a good job in replicating the results provided by factor models. In contrast, cross-sectional GLS $R^{2}$ statistics shown in Table 3 and Table 4 differ, given their dependency on errors resulting from time-series regressions used to estimate betas. In any case, Table 3 and Table 4 show that GLS $R^{2}$ statistics are lower than their OLS counterparts in all cases, which is consistent with the fact that the GLS $R^{2}$ statistic constitutes a more challenging hurdle for asset pricing models than the OLS $R^{2}$ statistic.

Regarding the HJ pricing kernel, Table 4 shows that it results in a perfect fit for all portfolios under consideration, providing OLS and GLS $R^{2}$ statistics equal to one, MAEs equal to zero, and *J*-tests that assign zero probability of rejection in all cases. Figure 1 plots the real and fitted values provided by all SDF-mimicking portfolios, with the angle of 45 degrees representing a perfect fit. As shown, all plots in Figure 1 are consistent with the results in Table 3 and Table 4, with the Fama–French three- and five-factor models providing the best results, the CCAPM and the CAPM performing significantly worse, and the HJ pricing kernel ensuring a perfect fit.

Table 4 also summarizes the main descriptive statistics for all SDF-mimicking portfolios, more particularly, their standard deviations, skewness, and kurtosis. It is worth noting that, since our test assets comprise solely excess returns, the mean of pricing kernels is set to one in order to allow comparability with the HJ pricing kernel, under the assumption that $R_{t+1}^{f}=1$ in Expression (18). Figure 2 depicts the histogram of standardized SDF-mimicking portfolios for all models under study. As shown in Table 4 and Figure 2, most pricing kernels share common features. Specifically, in most cases, the SDF-mimicking portfolios follow a skewed leptokurtic distribution, which is consistent with the high randomness required for pricing kernels to correctly price assets [4] (p. 95).

Noteworthy, although the HJ pricing kernel delivers $R^{2}$ statistics equal to one and a perfect fit for all portfolios, it exhibits a relatively low volatility. In contrast, the SDF-mimicking portfolios of the best-performing models—i.e., the Fama-French models—are extraordinarily volatile, which is coherent with the minimum-variance of the HJ pricing kernel, as defined by [14]. This fact emphasizes the importance of higher-order moments in evaluating the performance of asset pricing models and, consequently, the pertinence of entropy-based methodologies for that purpose.

Figure 3 plots the evolution over time of the demeaned SDF-mimicking portfolios and asset returns, for all models into consideration. Although all plots in Figure 3 are consistent with the stationary nature of the pricing kernel, the SDF-mimicking portfolios of the best-performing models do not appear to follow an independent and identically distributed (i.i.d.) pattern, especially for size-BE/ME and size-P/CF portfolios. Remarkably, the SDF-mimicking portfolios of the Fama-French models track the HJ pricing kernel behavior precisely for size-BE/ME portfolios, while they exhibit an inverse pattern for size-P/CF and industry portfolios, especially for years ranging from 1992 to 2000. As noted below, our entropy model correctly captures these patterns, allowing us to explicitly decompose the explanatory power of the asset pricing models into a pricing kernel component and a correlation component.

Table 4 shows the results provided by Expressions (20) and (21) for $I_{x^{*}}\left(\pi \;\|\;\theta \right)$ and $I_{\rho }\left(\upsilon \;\|\;\mu \right)$, respectively. As shown, in general, the better the performance of the asset pricing model, the lower the relative entropy of the SDF-mimicking portfolio, and vice versa. Specifically, Table 4 shows that higher GLS $R^{2}$ statistics are always accompanied by lower divergences between the SDF-mimicking portfolio and the HJ pricing kernel, for at least one of the components defined in Expressions (20) and (21). Importantly, those models with SDF-mimicking portfolios following inverse patterns relative to the HJ pricing kernel—as is the case of Fama-French models in rows B3, B4, C3 and C4 (see Figure 3)—exhibit a correlation entropy $I_{\rho }\left(\upsilon \;\|\;\mu \right)$ significantly higher than those models in which the SDF-mimicking portfolio correctly tracks the HJ pricing kernel (rows A3 and A4), even when the pricing kernel component $I_{x^{*}}\left(\pi \;\|\;\theta \right)$ is not affected by this fact and the GLS $R^{2}$ statistic remains largely unchanged. This suggests that our entropy model adequately captures the correlation structure of the pricing kernels.

In order to make explicit the relationship between the GLS $R^{2}$ statistic and the relative entropy of the pricing kernels, as measured by $I_{x^{*}}\left(\pi \;\|\;\theta \right)$ and $I_{\rho }\left(\upsilon \;\|\;\mu \right)$, Panel A in Table 5 shows the results provided by the regression and correlation analysis for these variables. Expression (19) shows that the relationship of expected excess returns with $\rho_{t,m,R^{e}}$ and $\sigma_{t}\left(m_{t+1}\right)$ is not additive but multiplicative. Accordingly, we perform all calculations both in levels and in logs. Additionally, Panels B and C in Table 5 show the same analysis for the case of the OLS $R^{2}$ statistic and the MAE, respectively.

Panel A in Table 5 shows that the GLS $R^{2}$ statistic and our entropy-based model are strongly related, especially when we use logarithms. In that case, the regression coefficients for $I_{x^{*}}\left(\pi \;\|\;\theta \right)$ and $I_{\rho }\left(\upsilon \;\|\;\mu \right)$ are strongly significant and the $R^{2}$ statistic amounts to 77.5%. These results worsen when we use levels rather than logs, which is consistent with the multiplicative nature of terms in Expression (19). Regarding correlations, Panel A in Table 5 shows that, while both $I_{x^{*}}\left(\pi \;\|\;\theta \right)$ and (especially) $I_{\rho }\left(\upsilon \;\|\;\mu \right)$ are strongly correlated with the GLS $R^{2}$ statistic, the correlation between $I_{x^{*}}\left(\pi \;\|\;\theta \right)$ and $I_{\rho }\left(\upsilon \;\|\;\mu \right)$ is small, which means that both terms are not redundant but complementary.

On the other hand, Panels B and C in Table 5 show that our entropy-based decomposition performs worse in explaining the OLS $R^{2}$ statistic and the MAE. In fact, while the regression coefficients shown in Panels B and C for $I_{x^{*}}\left(\pi \;\|\;\theta \right)$ remain statistically significant and their explanatory power even increases compared to that shown in Panel A for the GLS $R^{2}$ statistic, $I_{\rho }\left(\upsilon \;\|\;\mu \right)$ loses most of its significance and is far less correlated with the OLS $R^{2}$ statistic and the MAE than with the GLS $R^{2}$ statistic.

## 4. Discussion

The results shown in the previous section suggest that our relative entropy approach, as defined in Section 2, is strongly related to the mean-variance efficiency of the factor-mimicking portfolio of some of the most prominent asset pricing models in the literature, which makes it a useful tool to evaluate asset pricing model performance. Specifically, our results suggest that an entropy-based decomposition of the explanatory power of the model into a component that is purely determined by the SDF-mimicking portfolio—given by $I_{x^{*}}\left(\pi \;\|\;\theta \right)$—and a component determined by the correlation between the pricing kernel and asset returns—$I_{\rho }\left(\upsilon \;\|\;\mu \right)$—can contribute to improve the robustness and informativeness of model evaluation tests in an insightful manner. More particularly, our results offer three major findings on the relationship between the relative entropy of pricing kernels and the performance of asset pricing models. First, we show that our entropy-based decomposition is strongly related to the GLS $R^{2}$ statistic. Using data for the Australian stock market, our results show that the correlation between the GLS $R^{2}$ statistic and $I_{x^{*}}\left(\pi \;\|\;\theta \right)$ in logs is −47.6%, while that correlation amounts to −72.4% for $I_{\rho }\left(\upsilon \;\|\;\mu \right)$. Moreover, the $R^{2}$ statistic of the regression of the cross-sectional GLS $R^{2}$ statistic on $I_{x^{*}}\left(\pi \;\|\;\theta \right)$ and $I_{\rho }\left(\upsilon \;\|\;\mu \right)$ in logs amounts to 77.5%. These results suggest the existence of a strong negative relationship between the cross-sectional GLS $R^{2}$ statistic and the relative entropy of the pricing kernel, as defined in our study. We believe this is important evidence on the explanatory power of the model, given that the GLS $R^{2}$ statistic is directly influenced by the proximity of model factors to the minimum-variance frontier [8,21]. Furthermore, this statement is reinforced by the fact that our entropy-based-decomposition works worse in explaining both the OLS $R^{2}$ statistic and the MAE, as shown in Table 5.

Second, our results suggest that the relative entropy of the correlation between the pricing kernel and asset returns, $I_{\rho }\left(\upsilon \;\|\;\mu \right)$, is not only relevant in model evaluation, but it is more related to the proximity of model factors to the minimum-variance frontier than the pricing kernel component $I_{x^{*}}\left(\pi \;\|\;\theta \right)$. Indeed, Table 5 not only shows that the GLS $R^{2}$ statistic and $I_{\rho }\left(\upsilon \;\|\;\mu \right)$ are strongly correlated, but also that this correlation decreases significantly for other performance indicators less related to the mean-variance efficiency, such as the OLS $R^{2}$ statistic and the MAE. This is an important point, since most of the volatility and entropy bounds studied in the literature generally focus exclusively on the entropy of the pricing kernel. To the best of our knowledge, this is the first study to explicitly decompose the relative entropy of pricing kernels into the specific components defined above, using a SDF-mimicking portfolio approach for that purpose.

Third, our results provide proof that the correlation component $I_{\rho }\left(\upsilon \;\|\;\mu \right)$ allows the coherence of the pattern followed by pricing kernels over time to be accounted for, using the HJ pricing kernel as a reference. Given that the purpose of asset pricing theory is not only to produce models that result in small pricing errors, but also to understand the fundamental macroeconomic sources of risk that determine asset prices, this feature is crucial even for the best-performing models. Our results not only show that the higher the correlation component, the lower the explanatory power of the model, but also reveal that when the pricing kernels of the best-performing models follow well-differentiated patterns with respect to the HJ pricing kernel, their correlation component $I_{\rho }\left(\upsilon \;\|\;\mu \right)$ takes relatively high values, regardless of their GLS $R^{2}$ statistics. This feature is especially important for testing macroeconomic asset pricing models and conditional models, where date conventions (e.g., beginning-of-period or end-of-period timing assumptions) or the number of lags in instruments largely condition the correlations between the explanatory variables and asset payoffs [16,40].

These findings lead us to conclude that our entropy-based methodology can be useful to complement other fruitful studies on the area. Particularly, we believe that it can be a powerful tool for defining tighter bounds than those suggested by [9,20] for plausible pricing kernels. Indeed, our approach can allow models to split entropy bounds, as defined by these authors, into a first component purely determined by the randomness of the SDF, and a second component determined by the correlation between the pricing kernel and asset returns. This can help to more precisely diagnose the reasons that make some macroeconomic asset pricing models perform well with relatively low risk aversion coefficients, as is the case with the habit model of [41], or explain the reasons that make some multifactor models provide extremely low pricing errors, although the rationale behind their pricing kernels does not require defining an explicit economic setup, as is the case with Fama-French models. In this regard, the decomposition of the pricing kernels into an observable component and a potentially unobservable one, as suggested by [20], can particularly benefit from our entropy-based methodology.

## 5. Conclusions

Although research on asset pricing has provided us with a wide variety of models and theories aimed at finding the key determinants of asset prices and expected returns, the SDF or pricing kernel model is currently the dominant approach in asset pricing research. Moreover, since any valid pricing kernel is directly related to a mean-variance efficient return that carries all pricing information, discount factors, beta models, and mean-variance frontiers are distinct but closely related representations. As noted above, the fact that the law of one price guarantees the existence of at least one valid pricing kernel allows the SDF approach to produce a potentially infinite number of pricing kernels that mechanically price assets in sample. In this regard, the HJ pricing kernel, as defined by [14], lacks any economic content beyond its close relationship with the maximum Sharpe ratio available in the payoff space. However, this fact makes it a useful reference to evaluate the plausibility of candidate pricing kernels.

In this framework, entropy-based methodologies constitute an important step forward for that purpose. Pioneering this approach, [9] largely translates the volatility bound as defined by [14] into entropy terms. Although further research makes significant progress in determining the entropy components of pricing kernels and risk premiums [17,18,19,20], most of that research focuses on the relative entropy of the pricing kernel itself, ignoring the important role of the correlation between the pricing kernel and asset returns in evaluating the performance of asset pricing models.

Given the above considerations, it this paper we use the HJ pricing kernel as a reference to explicitly decompose the explanatory power of asset pricing models into two components, namely, the relative entropy of the candidate pricing kernel itself and the relative entropy of its correlation with asset returns. Our results show that, while frequently ignored by the literature on the topic, the correlation component is strongly related to the relative mean-variance efficiency of the model’s factor-mimicking portfolio, as measured by the cross-sectional GLS $R^{2}$ statistic. Conversely, the relative entropy of the pricing kernel itself is much more correlated with other measures less connected to the mean-variance frontier, such as the OLS $R^{2}$ statistic and the MAE. This opens a range of possibilities for the development of new testing procedures that better capture the dynamics of factor-mimicking portfolios, while avoiding the problems that arise from quadratic test statistics typically used to evaluate asset pricing model performance, such as the *J*-test for over-identifying restrictions or the GRS test statistic.

In this regard, we believe that a natural continuation of our research should address some specific issues, such as the derivation of an orthogonal decomposition of our relative entropy model, which allows us to determine the fraction of the explanatory power of the model that is attributable to both the pricing kernel and the correlation components. Moreover, it is convenient to explore the potential of our entropy-based decomposition to develop test statistics suitable for evaluating asset pricing models. For that purpose, approaches such as that suggested by [7] can be particularly useful. Additionally, reference portfolios other than the HJ pricing kernel should be validated. Entropy-based methods, such as those suggested by [42,43], can provide useful insight in this regard. Furthermore, given that the absence of arbitrage opportunities—that is, the fact that non-negative payoffs that are positive with positive probability have positive prices—guarantees the existence of at least a strictly positive pricing kernel [14] and the fact that intertemporal marginal rates of substitution must be positive, the implications of this constraint for our entropy-based decomposition must be sufficiently addressed. For this purpose, approaches that naturally impose the non-negativity restriction on the pricing kernel, such as that followed by [20], can be particularly useful.

In any case, future research should extend the scope of our paper to other asset pricing models, as well as to other markets and other assets, in order to check the validity of our methodology in different settings. Importantly, the use of payoffs other than excess returns implies that the expectation of the pricing kernel is meaningful for model evaluation, and the impact of this fact should be sufficiently addressed in future research.

## Figures and Tables

**Figure 1 entropy-22-00721-f001:**
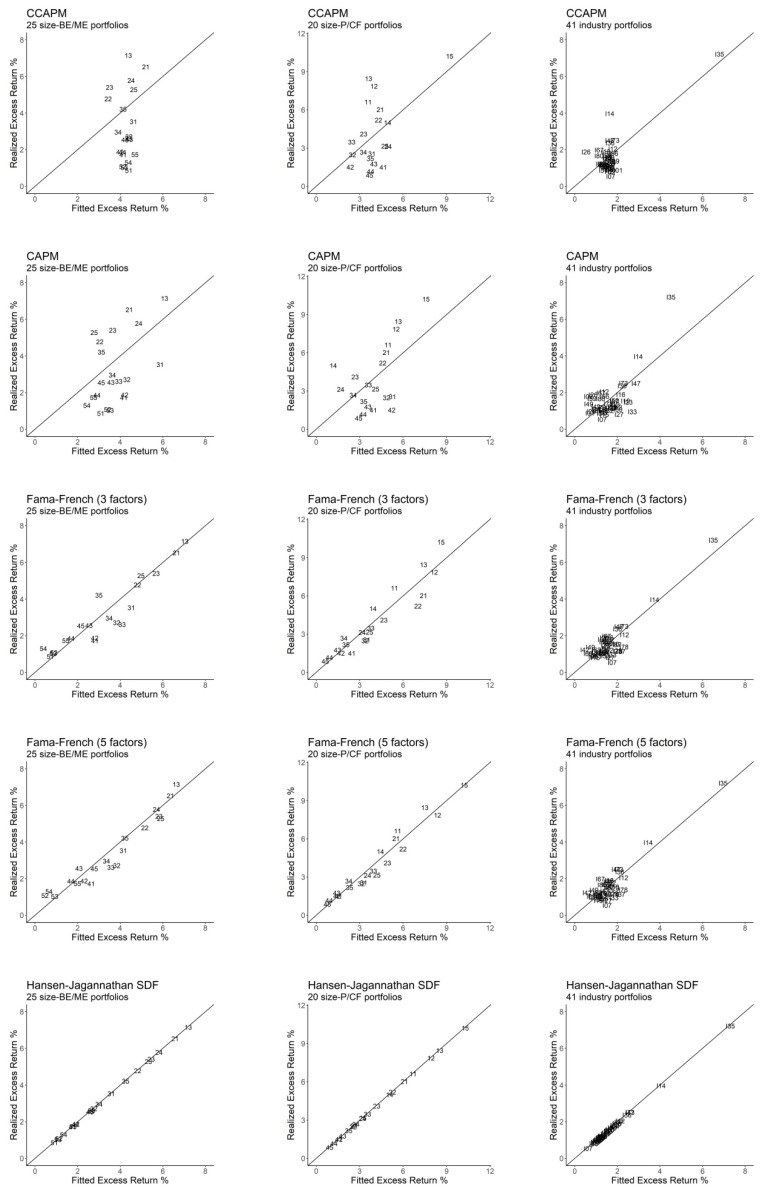
Realized excess returns versus fitted values. We depict the 25 portfolios size-book-to-market equity (BE/ME) according to a code with two numbers, the first number being the size code (with 1 being the smallest and 5 the largest) and the second number being the BE/ME ratio code (with 1 representing a low ratio and 5 a high ratio). Size-price-to-cash flow ratio (P/CF) portfolios are also represented by a code with two numbers, the first number being the size code (as in size-BE/ME portfolios, 1 is the smallest size and 5 the largest size) and the second number being the P/CF ratio code (with 1 representing a low ratio and 5 a high ratio). We depict industry portfolios according to the letter “I” followed by the first two digits of the SIC code.

**Figure 2 entropy-22-00721-f002:**
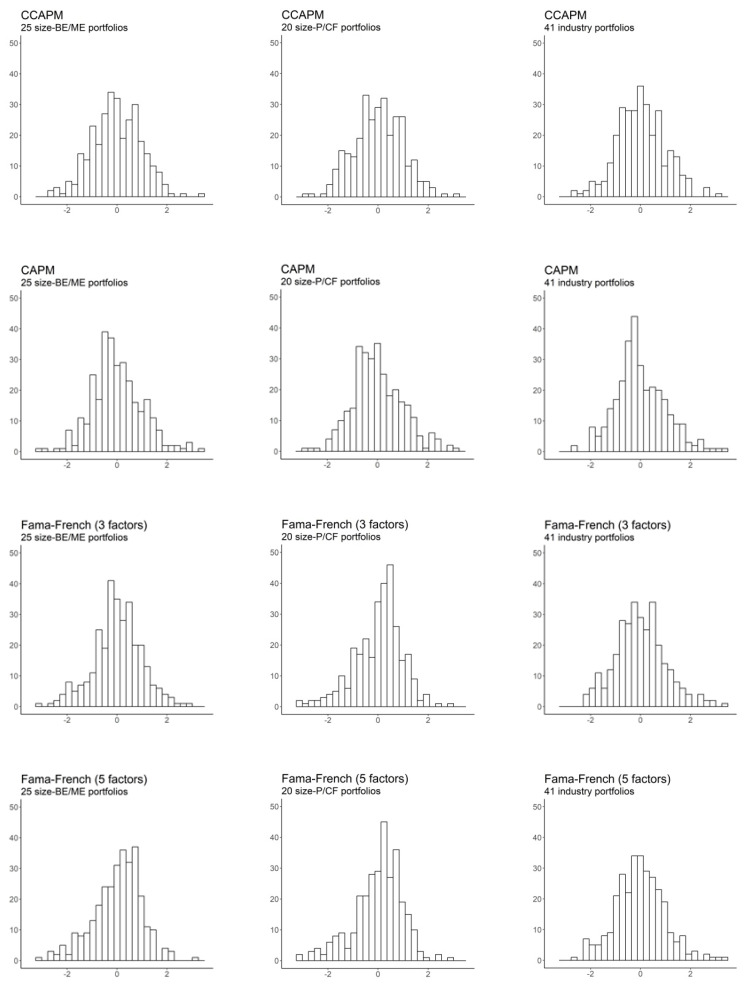
Histogram of standardized stochastic discount factor (SDF)-mimicking portfolios.

**Figure 3 entropy-22-00721-f003:**
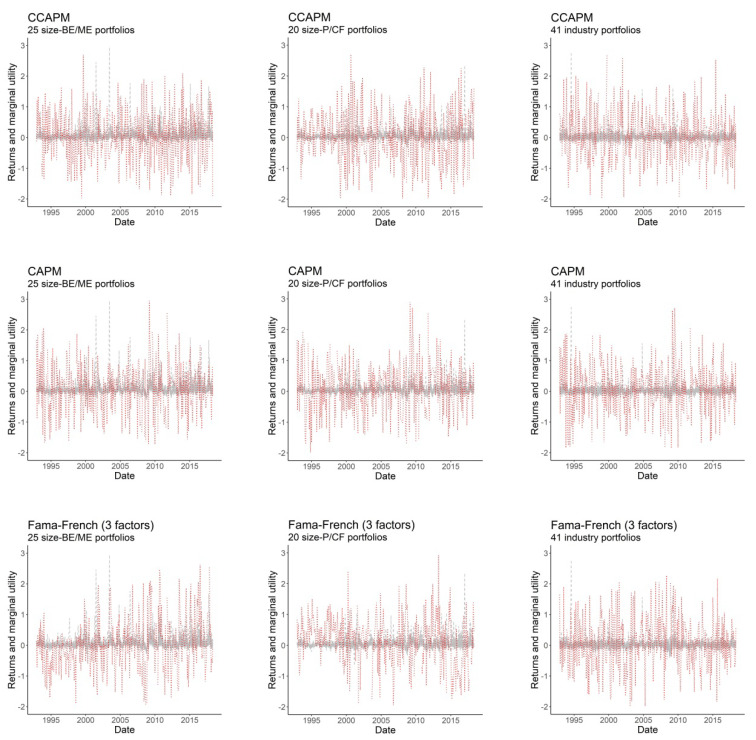
Excess returns and pricing kernels in time series. The figure depicts the evolution over time of the demeaned SDF-mimicking portfolios (dashed red lines) and the excess returns of portfolios under consideration (dashed grey lines).

**Table 1 entropy-22-00721-t001:** Summary statistics.

**Panel A: 25 Portfolios Size-BE/ME**											
	BE/ME quintiles						BE/ME quintiles				
Size	Low	2	3	4	High	Size	Low	2	3	4	High
	Means						St. Dev.				
Small	11.70	9.52	7.02	8.60	8.40	Small	32.81	20.41	16.84	20.30	13.90
2	6.40	4.65	5.27	5.65	5.15	2	20.28	12.38	13.18	15.70	10.36
3	3.40	2.59	2.48	2.83	4.08	3	9.80	8.87	8.53	7.41	7.48
4	1.59	1.74	2.43	1.73	2.41	4	7.39	6.71	6.30	5.27	6.28
Big	0.73	0.95	0.90	1.17	1.60	Big	5.15	4.53	4.78	4.22	6.38
**Panel B: 20 Portfolios Size-P/CF**											
	P/CF quintiles						P/CF quintiles				
Size	Low	2	3	4	High	Size	Low	2	3	4	High
	Means						St. Dev.				
Small	6.43	7.76	8.27	4.81	10.03	Small	13.43	15.87	14.92	12.90	55.04
2	5.83	5.01	3.92	2.94	2.96	2	13.14	11.56	8.44	7.51	8.98
3	2.37	2.29	3.28	2.49	1.99	3	8.74	8.59	8.85	5.18	6.05
Big	1.30	1.32	1.57	0.98	0.67	Big	7.53	9.02	7.41	4.34	4.13
**Panel C: 41 Industry Portfolios**											
	Mean deciles						St. Dev. Deciles				
	Low	2	3	4	5		Low	2	3	4	5
Deciles 1–5	0.75	0.89	0.96	1.05	1.10	Deciles 1–5	5.04	5.35	5.59	5.99	6.37
	6	7	8	9	High		6	7	8	9	High
Deciles 6-10	1.35	1.53	1.75	2.24	7.09	Deciles 6-10	6.97	7.49	8.80	11.12	89.35
**Panel D: Market Factors and Macroeconomic Series**											
	RMRF	SMB	HML	RMW	CMA		Δ*C* (quarters)		Δ*C* (years)		Δ*C* (Q4-Q4)
Means	1.09	4.19	0.52	1.48	0.38	Means	0.51	Means	1.75	Means	1.71
St. Dev.	3.99	6.78	5.63	5.43	5.00	St. Dev.	4.84	St. Dev.	1.56	St. Dev.	1.80

Note: All results are determined using monthly data, unless otherwise indicated. Means and standard deviations are percentages.

**Table 2 entropy-22-00721-t002:** Correlations.

**Panel A: 25 Portfolios Size-BE/ME**											
	BE/ME quintiles						BE/ME quintiles				
Size	Low	2	3	4	High	Size	Low	2	3	4	High
	RMRF						SMB				
Small	0.32	0.28	0.34	0.25	0.32	Small	0.33	0.52	0.47	0.45	0.61
2	0.23	0.31	0.32	0.32	0.35	2	0.51	0.57	0.57	0.61	0.60
3	0.57	0.52	0.52	0.57	0.52	3	0.49	0.53	0.57	0.55	0.49
4	0.62	0.68	0.66	0.71	0.62	4	0.41	0.44	0.45	0.35	0.37
Big	0.75	0.90	0.86	0.81	0.57	Big	0.08	0.11	0.06	0.02	0.18
	HML						RMW				
Small	−0.14	−0.20	−0.04	0.16	0.05	Small	0.13	0.20	0.33	0.25	0.36
2	−0.52	−0.14	−0.09	0.09	0.16	2	0.37	0.36	0.33	0.46	0.29
3	−0.20	−0.25	−0.09	−0.05	0.05	3	0.40	0.32	0.46	0.34	0.22
4	−0.19	−0.21	−0.19	−0.10	0.00	4	0.28	0.37	0.40	0.31	0.26
Big	−0.20	−0.18	−0.07	0.02	0.19	Big	0.26	0.17	0.10	0.08	0.17
	CMA						Δ*C* (quarters)				
Small	0.03	−0.05	−0.03	−0.04	−0.06	Small	0.06	−0.08	0.06	0.10	0.01
2	0.04	−0.07	−0.10	−0.33	−0.05	2	0.08	−0.06	−0.12	0.05	0.06
3	0.02	−0.07	−0.12	−0.15	0.10	3	0.09	0.04	0.02	−0.07	0.00
4	−0.07	−0.09	−0.10	−0.01	0.13	4	0.00	−0.04	0.05	−0.01	0.01
Big	−0.11	−0.14	0.05	0.06	0.13	Big	0.12	0.01	0.02	0.11	0.16
**Panel B: 20 Portfolios Size-P/CF**											
	P/CF quintiles						P/CF quintiles				
Size	Low	2	3	4	High	Size	Low	2	3	4	High
	RMRF						SMB				
Small	0.39	0.35	0.38	0.20	0.13	Small	0.50	0.65	0.62	0.34	0.18
2	0.39	0.43	0.43	0.39	0.51	2	0.68	0.75	0.65	0.46	0.49
3	0.62	0.59	0.48	0.68	0.66	3	0.46	0.46	0.49	0.36	0.35
Big	0.59	0.59	0.57	0.92	0.91	Big	0.32	0.17	0.17	0.13	0.03
	HML						RMW				
Small	−0.04	−0.07	0.01	0.09	−0.14	Small	0.34	0.39	0.23	0.13	0.07
2	−0.39	−0.08	−0.02	−0.02	−0.05	2	0.60	0.46	0.39	0.11	0.19
3	−0.22	−0.11	−0.03	−0.02	−0.16	3	0.39	0.35	0.38	0.23	0.24
Big	−0.16	−0.17	−0.04	−0.07	−0.11	Big	0.36	0.23	0.26	0.18	0.10
	CMA						Δ*C* (quarters)				
Small	0.03	0.00	−0.13	0.05	−0.03	Small	−0.01	0.03	0.04	0.08	0.07
2	−0.20	−0.17	0.06	−0.07	−0.09	2	0.01	0.03	−0.02	0.15	0.08
3	−0.08	−0.08	0.08	0.06	−0.05	3	0.04	−0.09	−0.08	−0.02	0.02
Big	−0.14	−0.06	0.07	−0.05	−0.03	Big	0.08	−0.05	0.01	0.05	0.05
**Panel C: 41 Industry Portfolios**											
	Correlation deciles										
	Low	2	3	4	5	6	7	8	9	10	Min
RMRF	0.29	0.40	0.44	0.48	0.52	0.57	0.61	0.64	0.68	0.77	0.08
SMB	−0.02	0.01	0.05	0.08	0.14	0.16	0.20	0.24	0.28	0.39	−0.11
HML	−0.14	−0.11	−0.09	−0.07	−0.05	−0.04	−0.03	−0.02	0.00	0.05	−0.19
RMW	0.01	0.03	0.04	0.07	0.10	0.11	0.13	0.17	0.24	0.34	−0.01
CMA	−0.06	−0.05	−0.05	−0.04	−0.02	−0.01	0.00	0.03	0.07	0.11	−0.13
Δ*C* (quarters)	−0.10	−0.03	0.00	0.04	0.07	0.08	0.11	0.12	0.17	0.24	−0.20

Note: All results are determined using quarterly data, unless otherwise indicated.

**Table 3 entropy-22-00721-t003:** Regression results for factor models.

			CCAPM	Fama-French Factors							
Row	Model	Intercept	*λ* _Δ*C*_	*λ_RMRF_*	*λ_SMB_*	*λ_HML_*	*λ_RMW_*	*λ_CMA_*	*R* ^2^	MAE (%)	*J*-Test
**Panel A: 25 Portfolios Size-BE/ME**											
A1	CCAPM	0.132	0.049						0.039	8.64	63.720
		(5.501)	(2.142)						0.015		(0.000)
A2	CAPM	−0.031		0.062					0.495	1.87	59.485
		(−1.695)		(3.543)					0.205		(0.000)
A3	Fama-French (3 factors)	−0.018		0.025	0.053	0.001			0.905	0.69	48.611
		(−1.241)		(1.632)	(9.682)	(0.139)			0.800		(0.001)
A4	Fama-French (5 factors)	−0.008		0.013	0.051	0.008	0.000	0.018	0.962	0.47	32.635
		(−.579)		(0.866)	(9.375)	(1.397)	(−0.032)	(1.719)	0.899		(0.026)
**Panel B: 20 Portfolios Size-P/CF**											
B1	CCAPM	0.105	0.130						0.321	6.26	13.599
		(2.389)	(0.993)						0.100		(0.755)
B2	CAPM	−0.027		0.057					0.324	1.87	58.275
		(−0.998)		(2.134)					0.023		(0.000)
B3	Fama-French (3 factors)	0.006		−0.001	0.056	−0.006			0.882	0.72	33.152
		(0.715)		(−0.067)	(6.925)	(−0.282)			0.784		(0.007)
B4	Fama-French (5 factors)	0.002		0.004	0.048	−0.017	−0.006	0.026	0.949	0.49	19.895
		(0.220)		(0.346)	(10.063)	(−0.638)	(−0.327)	(1.139)	0.906		(0.134)
**Panel C: 41 Industry Portfolios**											
C1	CCAPM	0.046	−0.035						0.624	1.63	57.663
		(4.042)	(−0.956)						0.374		(0.027)
C2	CAPM	−0.010		0.028					0.480	0.59	55.031
		(−0.584)		(1.289)					0.141		(0.046)
C3	Fama-French (3 factors)	−0.001		0.015	0.034	−0.054			0.786	0.41	35.219
		(−0.057)		(1.091)	(3.057)	(−0.803)			0.301		(0.553)
C4	Fama-French (5 factors)	0.003		0.009	0.027	−0.036	0.016	0.009	0.820	0.38	39.990
		(0.918)		(1.756)	(3.448)	(−1.477)	(1.433)	(0.333)	−0.835		(0.258)

Notes: The table displays two rows for each model, where the first row shows the coefficient estimates and the second row the *t*-statistics. Ordinary lest squares (OLS) and generalized least squares (GLS) $R^{2}$ statistics are provided, in that order. All *p*-values resulting from the *J*-tests are in parentheses. We use monthly data to estimate all models with the sole exception of the consumption-capital asset pricing model (CCAPM), which is estimated using quarterly data.

**Table 4 entropy-22-00721-t004:** Regression results for pricing kernel models.

								Mimicking Portfolio Statistics		
Row	Model	Intercept	*λ_x*_*	*R* ^2^	MAE (%)	*J*-Test	*I*·10^3^	St. Dev.	Skewness	Kurtosis
**Panel A: 25 Portfolios Size-BE/ME**										
A1	CCAPM	0.038	−0.065	0.020	2.48	198.447	34.088	0.26	0.17	3.64
		(9.665)	(−2.555)	−0.002		(0.000)	54.430			
A2	CAPM	−0.031	−2.377	0.495	1.87	60.753	0.013	1.54	0.31	3.62
		(−1.713)	(−3.582)	0.239		(0.000)	204.846			
A3	Fama-French (3 factors)	−0.018	−0.820	0.905	0.69	56.735	0.002	0.91	−0.36	3.84
		(−3.442)	(−8.746)	0.866		(0.000)	50.586			
A4	Fama-French (5 factors)	−0.008	−0.766	0.962	0.47	41.206	0.001	0.88	−0.65	3.90
		(−1.905)	(−9.102)	0.944		(0.011)	19.648			
A5	HJ pricing kernel	0.000	−0.733	1.000	0.00	0.000		0.86	−0.78	3.98
		(0.000)	(−9.394)	1.000		(1.000)				
**Panel B: 20 Portfolios Size-P/CF**										
B1	CCAPM	0.033	−0.281	0.293	1.78	136.998	0.004	0.51	−0.19	3.57
		(8.113)	(−2.159)	0.256		(0.000)	84.068			
B2	CAPM	−0.027	−1.993	0.324	1.87	59.771	0.011	1.41	0.35	3.36
		(−1.010)	(−2.162)	0.040		(0.000)	200.626			
B3	Fama-French (3 factors)	0.006	−0.551	0.882	0.72	40.265	0.002	0.74	−0.79	4.40
		(1.470)	(−6.198)	0.851		(0.002)	62.167			
B4	Fama-French (5 factors)	0.002	−0.558	0.949	0.49	29.279	0.001	0.75	−0.73	3.99
		(0.408)	(−5.173)	0.943		(0.045)	29.770			
B5	HJ pricing kernel	0.000	−0.635	1.000	0.00	0.000		0.80	−0.74	3.67
		(0.000)	(−5.226)	1.000		(1.000)				
**Panel C: 41 Industry Portfolios**										
C1	CCAPM	0.013	−0.100	0.617	0.50	75.236	0.002	0.34	−0.11	3.79
		(5.794)	(−1.090)	−0.280		(0.000)	1683.533			
C2	CAPM	−0.010	−0.476	0.480	0.59	55.926	0.003	0.69	0.43	3.42
		(−0.589)	(−1.300)	0.176		(0.039)	76.332			
C3	Fama-French (3 factors)	−0.001	−0.308	0.786	0.41	60.335	0.002	0.55	0.35	3.42
		(−0.054)	(−1.337)	0.616		(0.016)	83.164			
C4	Fama-French (5 factors)	0.003	−0.166	0.820	0.38	67.735	0.002	0.41	0.34	3.97
		(0.463)	(−1.353)	0.779		(0.003)	143.076			
C5	HJ pricing kernel	0.000	−0.392	1.000	0.00	0.000		0.63	0.06	3.12
		(0.000)	(−1.417)	1.000		(1.000)				

Notes: The table displays two rows for each model, where the first row shows the coefficient estimates and the second row the *t*-statistics. OLS and GLS $R^{2}$ statistics are provided, in that order. The column headed “*I*·10^3′^” shows the Kullback-Leibler divergences $I_{x^{*}}$ and $I_{\rho }$, in that order, in thousands. All *p*-values resulting from the *J*-tests are in parentheses. We use monthly data to estimate all models.

**Table 5 entropy-22-00721-t005:** Relationship between model performance and the relative entropy of pricing kernels.

**Panel A: GLS** $R^{2}$ **Statistic**						
	$\mathrm{GLS}\;R^{2}=a+bI_{x^{*}}+cI_{\rho }+\varepsilon$			$\mathrm{GLS}\;R^{2}=a+b\log\left(I_{x^{*}}\right)+c\log\left(I_{\rho }\right)+\varepsilon$		
	Estimate	Std. Error	*t* value	Estimate	Std. Error	*t* value
Intercept	0.635	0.112	5.655	−1.114	0.331	−3.367
$I_{x^{*}}$	−17.762	10.192	−1.743	−0.076	0.024	−3.167
$I_{\rho }$	−0.590	0.216	−2.725	−0.278	0.059	−4.682
$R^{2}$	0.513			0.775		
	Correlations: Levels			Correlations: Logs		
	$\mathrm{GLS}\;R^{2}$	$I_{x^{*}}$	$I_{\rho }$	$\mathrm{GLS}\;R^{2}$	$I_{x^{*}}$	$I_{\rho }$
$\mathrm{GLS}\;R^{2}$	1.000	−0.334	−0.591	1.000	−0.476	−0.724
$I_{x^{*}}$		1.000	−0.115		1.000	−0.034
$I_{\rho }$			1.000			1.000
**Panel B: OLS** $R^{2}$ **Statistic**						
	$\mathrm{OLS}\;R^{2}=a+bI_{x^{*}}+cI_{\rho }+\varepsilon$			$\mathrm{OLS}\;R^{2}=a+b\log\left(I_{x^{*}}\right)+c\log\left(I_{\rho }\right)+\varepsilon$		
	Estimate	Std. Error	*t* value	Estimate	Std. Error	*t* value
Intercept	0.704	0.089	7.952	−0.576	0.281	−2.047
$I_{x^{*}}$	−19.931	8.036	−2.480	−0.084	0.020	−4.128
$I_{\rho }$	−0.088	0.171	−0.514	−0.082	0.050	−1.619
$R^{2}$	0.408			0.681		
	Correlations: Levels			Correlations: Logs		
	$\mathrm{OLS}\;R^{2}$	$I_{x^{*}}$	$I_{\rho }$	$\mathrm{OLS}\;R^{2}$	$I_{x^{*}}$	$I_{\rho }$
$\mathrm{OLS}\;R^{2}$	1.000	−0.625	−0.059	1.000	−0.767	−0.278
$I_{x^{*}}$		1.000	−0.115		1.000	−0.034
$I_{\rho }$			1.000			1.000
**Panel C: MAE**						
	$\mathrm{MAE}=a+bI_{x^{*}}+cI_{\rho }+\varepsilon$			$\mathrm{MAE}=a+b\log\left(I_{x^{*}}\right)+c\log\left(I_{\rho }\right)+\varepsilon$		
	Estimate	Std. Error	*t* value	Estimate	Std. Error	*t* value
Intercept	0.009	0.002	4.188	−2.645	0.803	−3.292
$I_{x^{*}}$	0.459	0.200	2.291	0.169	0.058	2.911
$I_{\rho }$	−0.002	0.004	−0.359	0.056	0.144	0.391
$R^{2}$	0.385			0.488		
	Correlations: Levels			Correlations: Logs		
	MAE	$I_{x^{*}}$	$I_{\rho }$	MAE	$I_{x^{*}}$	$I_{\rho }$
MAE	1.000	0.614	−0.164	1.000	0.692	0.070
$I_{x^{*}}$		1.000	−0.115		1.000	−0.034
$I_{\rho }$			1.000			1.000

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
