# Peer review of "Relative Entropy and Minimum-Variance Pricing Kernel in Asset Pricing Model Evaluation"

_entropy, 2020, doi:10.3390/e22070721_

Round 1

Reviewer 1 Report

The authors provide an interesting paper.

Introduction is well written and quite straightforward.

The methodology is adequate and the subject is in a relatively new area.

The sample is adequate, to draw conclusions.

The results are well presented.

However, the conclusion is confusing.

Technical terms, in my opinion should not be included in the conclusion part. Technical terms can be moved in a separate discussion part or, better, in an appendix.

I would therefore feel it is essential for the authors to rewrite the conclusion by focusing on the significance of their findings and by excluding technical terms and symbols. Conclusion should be written by using simple terms and expressions.

Apart from that i am quite happy the paper to be published.

I therefore suggest acceptance of the paper, under the condition the authors will remove any technical terms and symbols from the conclusion.

Author Response

Response to reviewer #1

We are very grateful for the comments and constructive suggestions raised by the reviewer. We have reviewed them carefully and introduced the appropriate modifications in the manuscript (changes are marked in blue). Please, see below our detailed response. All page numbers refer to the revised version of the manuscript. The reviewer’s comments are in italics.

1. The authors provide an interesting paper.

Introduction is well written and quite straightforward.

The methodology is adequate and the subject is in a relatively new area.

The sample is adequate, to draw conclusions.

The results are well presented.

However, the conclusion is confusing.

Technical terms, in my opinion should not be included in the conclusion part. Technical terms can be moved in a separate discussion part or, better, in an appendix.

I would therefore feel it is essential for the authors to rewrite the conclusion by focusing on the significance of their findings and by excluding technical terms and symbols. Conclusion should be written by using simple terms and expressions.

Apart from that i am quite happy the paper to be published.

I therefore suggest acceptance of the paper, under the condition the authors will remove any technical terms and symbols from the conclusion.

Authors’ response:

We sincerely appreciate the constructive feedback from the reviewer and the suggestions provided to clarify the conclusions of the paper. Following the suggestion of the reviewer, we have completely rearranged the conclusions section. Specifically, we have moved the technical part to a separate discussion section (pages 17-18) and have rewritten the conclusions in order to promote clarity (pages 18-19).

Thus, the updated conclusions section begins by summarizing the general framework of the model and continues by emphasizing the significance of the findings, removing all symbols and minimizing technical terms and expressions. We have only preserved the part of the previous version related to future work, in order to comply with a recommendation suggested by another reviewer.

Reviewer 2 Report

Reviewer report for Entropy: “Relative entropy and minimum-variance pricing kernel in asset pricing model evaluation”

In this paper the author suggests an interesting approach to evaluating the performance of asset pricing models in a broad sense, which allows dividing their explanatory power into two components, namely: (1) the relative entropy of the pricing kernel, and (2) the relative entropy of the correlation between the pricing kernel and asset returns. The author uses the minimum-variance pricing kernel suggested by Jagannathan and Wang (1991) as a reference for that purpose. Although asset pricing literature emphasizes the benefits of entropy bounds for determining desirable properties for pricing kernels, the entropy decomposition suggested by the author complements previous research on the topic. The author uses different sorts of portfolios for the Australian stock market to study the relationship between the relative entropy of the pricing kernel, as defined in the paper, with other measures commonly used to assess the performance of asset pricing models. Paper results suggest that the relative entropy of the pricing kernel is strongly correlated with the R-squared of the cross-sectional regression of expected returns on betas, determined using the generalized least squares (GLS) method rather than ordinary least squares (OLS). This correlation is particularly high for the relative entropy of the correlation between the pricing kernel and asset returns.

Overall, I find the paper to be interesting and well-written. I think the research question is an important one, and the approach suggested makes sense in the light of the current body of knowledge. Nevertheless, I have some suggestions that should be addressed in the paper. I share these aspects below:

  1. The author explicitly relates his/her entropy model to the GLS R-squared statistic that results from the regression of expected returns on betas. However, the OLS R-squared is by far more widely used in the asset pricing literature. The author properly justifies this decision by citing Lewellen, Nagel and Shanken (2010), who provide some important recommendations for improving testing procedures in asset pricing. Specifically, in lines 88-97 the author states that:

while the cross-sectional OLS R2 statistic is one of the most widely used tool for evaluating model performance, it has little connection to the factors’ proximity to the minimum-variance frontier in the mean-variance space of returns […] this problem is significantly mitigated by the cross-sectional GLS R2 statistic. […] the GLS R2 statistic is entirely determined by the factor’s proximity to the minimum-variance frontier, which makes it more suitable than the OLS R2 statistic to evaluate the performance of asset pricing models”.

Results in Table 5 relate the relative entropy of the pricing kernel to the GLS R-squared, but it completely ignores the OLS R-squared statistic. In order to provide stronger evidence on the relationship between the relative entropy of the pricing kernel and the proximity of model factors to the minimum-variance frontier, the author should extend his/her analysis to the OLS R-squared in Table 5.

  1. In line with the previous suggestion, it would be also interesting that the author examines the relationship between the relative entropy of the pricing kernel and the mean absolute error (MAE) of models being considered. Although the MAE is an intuitive and easy-tractable indicator for asset pricing performance, its connection to the proximity of model factors to the minimum-variance frontier is often spurious. Therefore, for the purpose of the study it is appropriate to examine whether the MAE and the relative entropy of the pricing kernel are related.
  2. Although the author leaves for future work the use of pricing kernels other than the minimum-variance stochastic discount factor as a reference in his/her entropy model (lines 492-494), the importance of extending this analysis to strictly positive pricing kernels has not been sufficiently addressed. This point is crucial given its direct implications on macroeconomic explanations for asset prices, such as those provided by the utility theory. The author should make explicit reference to this point in the discussion and conclusions Section.

REFERENCES

Jagannathan, R., & Wang, Z. (1996). The Conditional CAPM and the Cross-Section of Expected Returns. The Journal of Finance, 51(1), 3-53.

Lewellen, J., Nagel, S. and Shanken, J. (2010)- A Skeptical Appraisal of Asset Pricing Tests. Journal of Financial Economics, 96 (2), 175-194.

Author Response

Response to reviewer #2

We are very grateful for the comments and questions raised by the reviewer. We have reviewed them all carefully and introduced the appropriate modifications in the manuscript when necessary (changes are marked in green). Please, see below our detailed response to comments. All line numbers refer to the revised version of the manuscript. The reviewer’s comments are in italics.

  1. The author explicitly relates his/her entropy model to the GLS R-squared statistic that results from the regression of expected returns on betas. However, the OLS R-squared is by far more widely used in the asset pricing literature. The author properly justifies this decision by citing Lewellen, Nagel and Shanken (2010), who provide some important recommendations for improving testing procedures in asset pricing. Specifically, in lines 88-97 the author states that:

 “while the cross-sectional OLS R2 statistic is one of the most widely used tool for evaluating model performance, it has little connection to the factors’ proximity to the minimum-variance frontier in the mean-variance space of returns […] this problem is significantly mitigated by the cross-sectional GLS R2 statistic. […] the GLS R2 statistic is entirely determined by the factor’s proximity to the minimum-variance frontier, which makes it more suitable than the OLS R2 statistic to evaluate the performance of asset pricing models”.

 Results in Table 5 relate the relative entropy of the pricing kernel to the GLS R-squared, but it completely ignores the OLS R-squared statistic. In order to provide stronger evidence on the relationship between the relative entropy of the pricing kernel and the proximity of model factors to the minimum-variance frontier, the author should extend his/her analysis to the OLS R-squared in Table 5.

Authors’ response:

We really appreciate the constructive feedback from the reviewer and the suggestion provided to deliver stronger evidence on the relationship between the relative entropy of the pricing kernel and the proximity of model factors to the minimum-variance frontier. Following the suggestion of the reviewer, we have extended our analysis to study the relationship between our entropy-based decomposition and the OLS  statistic. For that purpose, we have rearranged Table 5 into three panels, with Panel A showing the results of the regression and correlation analysis for the cross-sectional GLS  statistic, and Panels B and C showing the results for the OLS  statistic and the MAE, respectively (see below the response to comment number 2). Additionally, we have added lines 418-420, 430-435, 455-457 and 460-463 in the manuscript to explain the implications of the new calculations.

As we explain in the manuscript, these new results strongly complement those shown in the previous version of the paper. In particular, Table 5 shows that the cross-sectional GLS  statistic is more closely related to our entropy-based decomposition than the OLS  statistic and the MAE. However, while the relative entropy of the pricing kernel itself, , is significantly correlated with both the OLS  statistic and the MAE, the relative entropy of the correlation between the pricing kernel and asset returns, , is strongly correlated with the GLS  statistic. This suggests that it is this second component that best captures the proximity of model factors to the minimum-variance frontier. In our opinion, this is an important result, since most of the volatility and entropy bounds studied in the literature generally focus exclusively on the entropy of the pricing kernel. We thank the reviewer for this useful suggestion that clearly reinforces the conclusion of our paper.

  1. In line with the previous suggestion, it would be also interesting that the author examines the relationship between the relative entropy of the pricing kernel and the mean absolute error (MAE) of models being considered. Although the MAE is an intuitive and easy-tractable indicator for asset pricing performance, its connection to the proximity of model factors to the minimum-variance frontier is often spurious. Therefore, for the purpose of the study it is appropriate to examine whether the MAE and the relative entropy of the pricing kernel are related.

Authors’ response:

As noted above in response to comment number 1, we have redone Table 5 to include the regression and correlation analysis for the MAE (see Panel C). This analysis together with that for the OLS  statistic allows us to conclude that the relative entropy of the correlation between the pricing kernel and asset returns, , is strongly related to the GLS  statistic, but this relationship is much weaker for other performance indicators less related to the mean-variance efficiency, such as the OLS  statistic and the MAE.

  1. Although the author leaves for future work the use of pricing kernels other than the minimum-variance stochastic discount factor as a reference in his/her entropy model (lines 492-494), the importance of extending this analysis to strictly positive pricing kernels has not been sufficiently addressed. This point is crucial given its direct implications on macroeconomic explanations for asset prices, such as those provided by the utility theory. The author should make explicit reference to this point in the discussion and conclusions Section.

Authors’ response:

As suggested by the reviewer, in lines 531-537 we have made explicit reference to the importance of deepening our understanding of the implications of strictly positive pricing kernels for our entropy-based decomposition, in order to strengthen consistence between discount rates and intertemporal marginal rates of substitution.